# Enamel Matrix Derivative and Autogenous Bone Graft for Periodontal Regeneration of Intrabony Defects in Humans: A Systematic Review and Meta-Analysis

**DOI:** 10.3390/ma12162634

**Published:** 2019-08-19

**Authors:** Marco Annunziata, Angelantonio Piccirillo, Francesco Perillo, Gennaro Cecoro, Livia Nastri, Luigi Guida

**Affiliations:** Multidisciplinary Department of Medical-Surgical and Dental Specialties, University of Campania “Luigi Vanvitelli”, 80138 Naples, Italy

**Keywords:** intrabony defect, periodontal disease, enamel matrix derivative, autogenous bone, periodontal regeneration

## Abstract

The combination of enamel matrix derivative (EMD) with an autogenous bone graft in periodontal regeneration has been proposed to improve clinical outcomes, especially in case of deep non-contained periodontal defects, with variable results. The aim of the present systematic review and meta-analysis was to assess the efficacy of EMD in combination with autogenous bone graft compared with the use of EMD alone for the regeneration of periodontal intrabony defects. A literature search in PubMed and in the Cochrane Central Register of Controlled Trials was carried out on February 2019 using an ad-hoc search string created by two independent and calibrated reviewers. All randomized controlled trials (RCTs) comparing a combination of EMD and autogenous bone graft with EMD alone for the treatment of periodontal intrabony defects were included. Studies involving other graft materials were excluded. The requested follow-up was at least 6 months. There was no restriction on age or number of patients. Standard difference in means between test and control groups as well as relative forest plots were calculated for clinical attachment level gain (CALgain), probing depth reduction (PDred), and gingival recession increase (RECinc). Three RCTs reporting on 79 patients and 98 intrabony defects were selected for the analysis. Statistical heterogeneity was detected as significantly high in the analysis of PDred and RECinc (I^2^ = 85.28%, *p* = 0.001; I^2^ = 73.95%, *p* = 0.022, respectively), but not in the analysis of CALgain (I^2^ = 59.30%, *p* = 0.086). Standard difference in means (SDM) for CALgain between test and control groups amounted to −0.34 mm (95% CI −0.77 to 0.09; *p* = 0.12). SDM for PDred amounted to −0.43 mm (95% CI −0.86 to 0.01; *p* = 0.06). SDM for RECinc amounted to 0.12 mm (95% CI −0.30 to 0.55. *p* = 0.57). Within their limits, the obtained results indicate that the combination of enamel matrix derivative and autogenous bone graft may result in non-significant additional clinical improvements in terms of CALgain, PDred, and RECinc compared with those obtained with EMD alone. Several factors, including the surgical protocol used (e.g. supracrestal soft tissue preservation techniques) could have masked the potential additional benefit of the combined approach. Further well-designed randomized controlled trials, with well-defined selection criteria and operative protocols, are needed to draw more definite conclusions.

## 1. Introduction

Periodontitis is a multifactorial, chronic, infective disease of the periodontal tissues that affect human populations worldwide, characterized by an inflammatory response of the periodontal tissues to periodontal pathogenic bacteria [1]. Risk factors are oral hygiene, diabetes, smoking, genetic predisposition, and lack of dental visits. Periodontitis is characterized by periodontal breakdown with apical migration of the junctional epithelium, clinical attachment loss and bone loss that can induce horizontal and/or vertical bone defect formation. Vertical intrabony defects, also known as angular defects, can be treated by surgical procedures able to regenerate the lost tissues. 

After the motivation to oral hygiene and non-surgical therapy, which represent the starting point in periodontitis treatment, a re-evaluation of the patient’s condition to verify the reduction of periodontal inflammation and to plan, if necessary, a surgical approach is mandatory. The aim of the regenerative treatment of the periodontal intrabony defects is to obtain a new periodontal attachment with new cementum, periodontal ligament, and alveolar bone [2].

Guided tissue regeneration (GTR) is based on the placement of non-resorbable or bio-resorbable membranes in order to create a barrier effect protecting against epithelial and connective apical migration. Furthermore, these membranes provide a tent effect in order to maintain the space between the bone and the root surface and to enable repopulation of periodontal ligament, cementum, and alveolar bone [3]. GTR can be combined with a biomaterial graft in case of non-self-supporting intrabony defects [4].

Another technique to achieve the regeneration of destroyed periodontal tissues is induced tissue regeneration (ITR). ITR is based on the use of enamel matrix derivative (EMD) mainly composed of amelogenins—a family of hydrophobic porcine tooth-derived proteins. EMD is demonstrated to have a significant role in the behavior of several cell populations, in terms of cell proliferation, survival, adhesion, and release of growth factors, cytokines, and other molecules involved in periodontal and bone healing [5,6]. The proteins contained in EMD are able to induce the genesis of cementum and periodontal ligament during tooth formation, although the exact mechanism how EMD participates in the periodontal regeneration process is still unclear [7]. This protein can be used alone in periodontal regeneration, even if the EMD’s gel-like consistency limits its potential, especially in non-self-supporting defects. To overcome this limitation, a combined approach based on EMD with different biomaterial grafts has been proposed [8,9,10,11,12,13,14,15].

Autogenous bone (AB) is the most biocompatible graft biomaterial, and avoids the risk of immunologic reaction or disease transmission. It has an osteoconductive effect, providing a scaffold for osteoblasts to produce new bone and may also have an osteogenic effect promoting the proliferation and differentiation of osteoprogenitor cells [16,17]. 

A combined approach of EMD and AB, in the form of cortical particles, has been proposed [15,18] to promote regeneration in non-self-supporting intrabony defects. While AB grafts avoid flap collapse, overcoming the limits imposed by the gel consistency of EMD, and provides an osteoconductive effect, EMD induces the development of new cementum and periodontal ligament.

The clinical data about EMD in combination with autogenous bone are still limited, and the potential of this association needs to be further investigated. The purpose of this review and successive meta-analysis is to verify the clinical efficacy of EMD and autogenous bone compared with EMD alone in the regenerative periodontal surgery of periodontal intrabony defects.

## 2. Materials and Methods

This systematic review was prepared following the PRISMA (Preferred Reporting Items for Systematic Reviews and Meta-Analyses) guidelines (www.prisma-statement.org) [19].

### 2.1. Focused Question

The focused question was formulated according to the PICO (population, intervention, control, outcome) principle for evidence-based practice [20]: “In patients with intrabony defects, what is the clinical benefit of using the enamel matrix derivative (EMD) in conjunction with autogenous bone compared with EMD alone in terms of periodontal indices change?”

### 2.2. Search Strategy

A literature search was carried out on February 2019 by two independent and calibrated reviewers in the database of the National Library of Medicine MEDLINE/PubMed, in the Cochrane Central Register of Controlled Trials, and in the ClinicalTrials.gov website. The authors created and adopted an ad-hoc search string: “(autogenous bone OR autologous bone OR bone graft OR bone) AND (enamel protein OR enamel matrix protein derivative OR enamel matrix derivative OR dental enamel proteins OR emdogain OR EMG OR EMD) AND (intrabony defects OR intra bony defect OR infrabony defects OR infra bony defect OR regenerative periodontal treatment OR periodontal regeneration OR periodontal pocket surgery OR surgical flap)”.

### 2.3. Inclusion Criteria

The studies were included on the basis of the following criteria:English language.Randomized controlled trials (RCTs) comparing a combination of EMD and autogenous bone graft with the EMD alone for the treatment of periodontal intrabony defects.Studies including patients with advanced chronic or aggressive periodontitis with the presence of at least one intrabony defect with a probing depth of at least 6 mm and an intrabony component of at least 3 mm as detected on the radiographs.Studies with at least 6-month follow-up after surgery for the radiographic and clinical evaluation.

### 2.4. Exclusion Criteria

The studies were excluded on the basis of the following criteria:Studies not reporting clinical/radiographical data.Studies that considered the use of EMD in combination with other biomaterials.Studies comparing the use of EMD in combination with autogenous bone graft with open-flap debridement, guided tissue regeneration, or autogenous bone graft alone.Preclinical studies, case series, case reports, retrospective studies, letters to the editor, technical reports, narrative reviews, conference abstracts.

### 2.5. Data Extraction and Analysis

Two independent reviewers (M.A., A.P.) screened the titles identified by the search. The abstracts were obtained for studies of possible relevance. For abstracts meeting the eligibility criteria or not providing sufficient data, the full texts were carefully read and analyzed for inclusion and data extraction. The inter-examiner agreement was verified by kappa coefficient, and any discrepancy resolved via discussion.

### 2.6. Outcome Measures

The primary outcome measures (i.e., true endpoint outcome) included:Change in clinical attachment level (CAL) or relative attachment level (RAL).

The secondary outcome measures (i.e., surrogate endpoint outcomes) included:Change in probing depth (PD);Change in gingival recession (REC).

### 2.7. Methodological Quality Assessment

The methodological quality of each study was assessed according to the criteria suggested by Van der Weijden et al. [21], with some modifications. The potential risk of bias was calculated based on the quality criteria met by each study.

### 2.8. Data Analysis

After analysis of the selected studies, data on clinical, intrasurgical, and radiographical study outcomes were collected by two independent reviewers (M.A., A.P.). Means/medians and their standard deviations/errors were recorded when available, and a quantitative synthesis by a meta-analysis was performed.

Dedicated software was used for data analysis (Comprehensive Meta-Analysis, Biostat, NJ 07631 USA). Mean differences and 95% confidence intervals of differences (95% CI) were calculated for PD, CAL, and REC. Statistical heterogeneity was verified by the I^2^ test, with *p*-values below 0.05 considered significant. Both fixed and random effect models were used. Forest plots were utilized to illustrate the weighted mean of the outcome in each study and the final estimate.

## 3. Results

From the initial search, 590 items in MEDLINE/PubMed and 27 items from other sources (i.e., the Cochrane Central Register of Controlled Trials and the ClinicalTrials.gov website) were found. After duplicates and items with no data available were removed, 591 records remained. After screening of titles and abstracts for inclusion/exclusion criteria, 588 studies were excluded. At the end of the process, three randomized controlled trials with parallel design [18,22,23] published between 2007 and 2016 were included in this systematic review (Figure 1). The inter-examiner kappa coefficient was 0.93. A list of the excluded studies with the reason of exclusion is available as Appendix A.

The number of participants ranged from 12 [23] to 40 [22]. There were 98 intrabony defects in 79 patients, with an age range between 30 and 65 years, and 54.4% (n = 43/79) were males.

Regarding tooth type and location, all of the examined studies included all types of teeth (incisors, canines, premolars, and molars) of maxilla and mandible. Regarding defect type, in the work of Guida et al. [18], the authors selected only sites with predominantly one- or two-wall component. In the study of Yilmaz et al. [22], the authors evaluated two- and three-wall intrabony periodontal defects. In the study of Agrali et al. [23] the authors included one, one–two, and one–two–three-walled defects.

The main characteristics of selected studies are described in Table 1, Table 2 and Table 3. In particular, design, characteristics of the study population, defect localization, number of defect walls, and type of bone harvested are explained in Table 1.

Aim, inclusion criteria, and surgical protocol are reported in Table 2.

Follow-up, outcome measures, methods of evaluation of the use of EMD in conjunction with autogenous bone, and conclusions are described in Table 3.

The results of the methodological quality assessment of the included studies revealed a low estimated potential risk of bias for all three studies (Table 4). All three studies investigated the effectiveness of a regenerative procedure based on the use of EMD in combination with autogenous bone graft.

Clinical parameters such as probing depth (PD) clinical attachment level (CAL) or relative attachment level (RAL), and gingival recession (REC) were evaluated in all studies. Plaque index (PI), gingival index (GI), and bleeding on probing (BOP) were evaluated in two studies [21,22]. Local plaque score (LPS) and local bleeding score (LBS) were recorded dichotomously in one study [18]. Radiographic parameters such as depth of the defect (DEPTH) and the radiographic defect angle (ANGLE) were measured in one study [18]. The radiographic bone fill percentage was evaluated in two studies [17,22].

Intrasurgical measurement such as the intra-bony component of the defect (IBD), intrabony defect depth (IDD), and the depth of the intrabony component (INTRA) were measured in all three studies [17,21,22]. Probing bone level (PBL), as the distance from the cemento-enamel junction to the apical end of the defect, was evaluated in two studies [17,21].

Moreover, in one study [23], the authors assessed the gingival crevicular fluid transforming growth factor-β1 levels by the collection of samples of gingival crevicular fluid.

In Table 5 and Table 6 we explain the clinical, radiographical, and intrasurgical characteristics of intrabony defects at baseline of the included studies. Changes in BOP, PD, CAL, REC, RAL, DEPTH, and bone fill at last follow-up are described in Table 7.

The standardized average differences between the test and control groups were calculated, and the relative forest plots were realized (Figure 2, Figure 3 and Figure 4) in terms of gain of clinical attachment level (CALgain), reduction of probing depth (PDred), and increase in gum recession (RECinc).

Statistical heterogeneity was detected as significantly high in the analysis of PDred and RECinc (I^2^ = 85.28%, *p* = 0.001; I^2^ = 73.95%, *p* = 0.022, respectively), but not in the analysis of CALgain (I^2^ = 59.30%, *p* = 0.086). Standard difference in means (SDM) for CALgain between test and control groups (fixed model) amounted to −0.34 mm (95% CI −0.77 to 0.09; *p* = 0.12). SDM for PDred amounted to −0.43 mm (95% CI −0.86 to 0.01; *p* = 0.06). SDM for RECinc amounted to 0.12 mm (95% CI −0.30 to 0.55; *p* = 0.57).

## 4. Discussion

The present systematic review assessed the efficacy of the use of EMD in combination with autogenous bone grafts compared with the use of EMD alone in the treatment of periodontal intrabony defects based on existing RCTs. The obtained results from the included studies indicate how this surgical approach has been investigated only by a very low number of well-designed clinical studies. The evaluation period of 6 to 12 months was selected because this is the follow-up time used in most clinical studies to evaluate the outcomes of regenerative periodontal surgery.

In order to improve the clinical results obtained with EMD and to overcome the flap collapse that occurs after surgery in the non-self-supporting intrabony defects leading to a limitation of the space available for regeneration, the combination of EMD with different types of grafting materials have been proposed. The main part of these studies investigate the combination of EMD and several biomaterials compared with the use of EMD alone in the treatment of periodontal intrabony defects [24]. Very few studies instead consider the use of EMD in combination with autogenous bone, and no systematic reviews specifically focused on this topic have been published in the literature.

Studies assessing the effect of the conjunction of EMD with other biomaterials indicate that the combination of EMD and bone grafts may result in additional clinical improvements in terms of CALgain and PD reduction compared with those obtained with EMD alone. However, the potential influence of the chosen graft material or the surgical procedure (i.e., flap design) on the clinical outcomes is unclear [24]. 

Autogenous bone grafting involves the harvesting of bone obtained from the same individual receiving the graft. Commonly, this bone is harvested from the mandibular ramus or the mandibular symphysis from a second surgical site, different from the receiving site. When a lower graft volume is needed, cortical bone particles can be obtained from a donor site adjacent to the receiving one, using dedicated harvesting devices (bone scrapers). The advantages of the autogenous bone graft there is its ability to be used as block or particulate, as well as its osteoconductive, osteoinductive and osteogenic potential. Conversely, limitations of the autogenous bone graft is its unpredictable resorption and the increased morbidity due to the donor site [16,17,25].

All three studies included here presented a high methodological quality. However, a substantial heterogeneity of study characteristics among them was found in terms of patient population, defect localization, number of defect walls, surgical protocol (e.g., supracrestal soft tissue preservation techniques), outcome variables, and follow-up time. These differences among studies made the data analysis more difficult.

All three works mainly concern defects with one or two walls, even if the work of Yilmaz et al. [22] included a higher percentage of three-wall pockets. The RCTs of Guida et al. [18] and Agrali et al. [23] showed no significant differences in favor of EMD + AB compared to EMD alone. These differences were instead significant in favor of the combination therapy in the work of Yilmaz et al. [22]. Thus, EMD alone also worked well in pockets with one and two walls, and the addition of autogenous bone did not produce significant differences for any of the considered outcomes. Only one study [18] described a statistically significant reduction of REC increase in favor of EMD + autogenous bone compared to EMD alone.

In carrying out the systematic review, we also analyzed two case series [15,26] in which the authors selected only sites with predominantly one- or two-wall components treated with EMD + autogenous bone. We excluded these works on the basis of the inclusion/exclusion criteria. As for the study of Trombelli et al. [15], the results showed statistically significant differences in terms of PDred and CALgain, and no statistically significant differences in terms of REC increase. Regarding the case series of Ferrarotti et al. [26], the results showed statistically significant differences in terms of PDred when the intrabony component of the defect (INFRA) was ≥5 mm, CALgain, DEPTH/INFRA reduction (greater entity when INFRA ≥ 5mm), and no significant differences in terms of REC increase.

## 5. Conclusions

Within their limits, the obtained results from the high-quality studies included indicate that the combination of enamel matrix derivative and autogenous bone graft result in non-significant additional clinical improvements in terms of CALgain, PDred, and RECinc compared with those obtained with EMD alone. The use of EMD alone allows for a more manageable and less-invasive treatment. Several factors, including the surgical protocol used (e.g., supracrestal soft tissue preservation techniques) could have masked the potential additional benefit of the combined approach. Further well-designed randomized controlled trials, with well-defined selection criteria and operative protocols, are needed to draw more definite conclusions.

## Figures and Tables

**Figure 1 materials-12-02634-f001:**
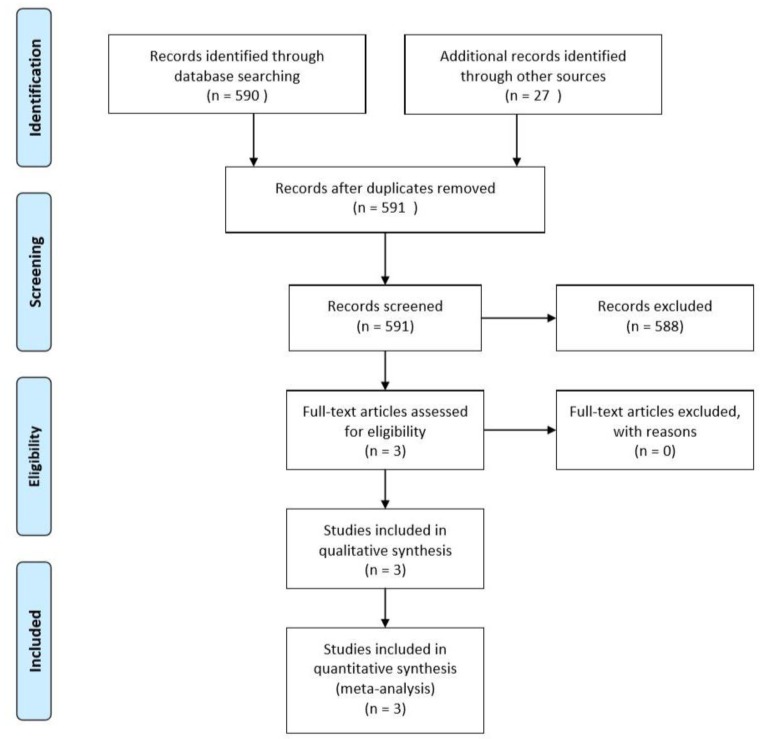
Flow diagram (PRISMA format) of the screening and selection process.

**Figure 2 materials-12-02634-f002:**
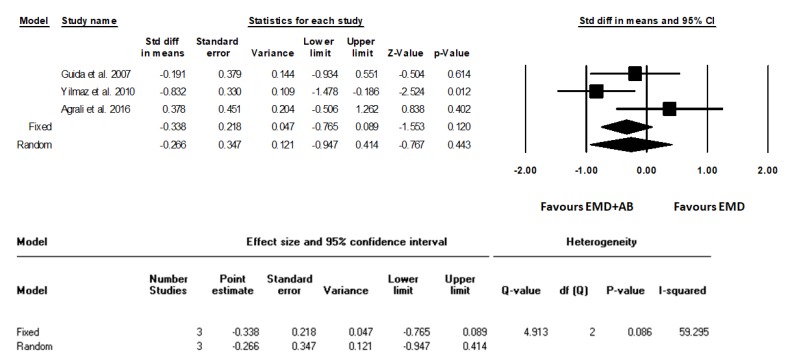
Forest plot from fixed and random effects of meta-analysis evaluating the differences in gain in clinical attachment level (CALgain, mm) after surgical treatment using EMD and autogenous bone or EMD alone (weighted mean difference, 95% CI) with the related heterogeneity analysis.

**Figure 3 materials-12-02634-f003:**
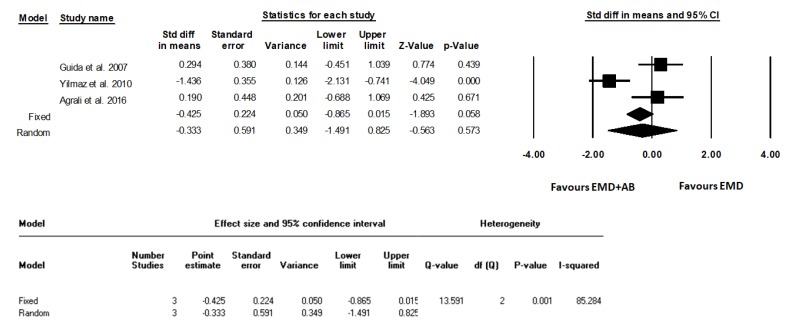
Forest plot from fixed and random effects of meta-analysis evaluating the differences in reduction of probing depth (PDred, mm) after surgical treatment using EMD and autogenous bone or EMD alone (weighted mean difference, 95% CI) with the related heterogeneity analysis.

**Figure 4 materials-12-02634-f004:**
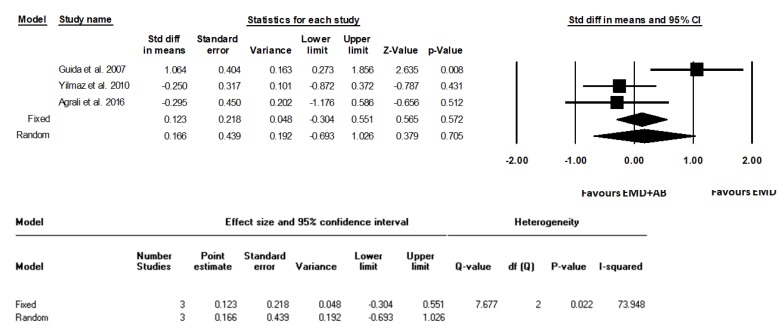
Forest plot from fixed and random effects of meta-analysis evaluating the differences in increase in gum recession (RECinc, mm) after surgical treatment using EMD and autogenous bone or EMD alone (weighted mean difference, 95% CI) with the related heterogeneity analysis.

**Table 1 materials-12-02634-t001:** Main characteristics of the selected studies: study population.

Study	Design	Patients/Defects Gender (M/F)	Mean Age (Range or ±SD)	Study Groups (Patients/Defects)		Defect Localization	Number of Defect Walls	Type of Bone Harvested
Guida et al., 2007 [18]	RCT (Pa)	27/28 13/14	46.3 ± 8.7 (30–65 years)	Test EMD + AB (14)	Ctr EMD (14)	Max: 13 def. (7 EMD + AB, 6 EMD) Mdb: 15 def. (7 EMD + AB, 8 EMD) IN/CA: 12 def. (5 EMD + AB, 7 EMD) PM/MO: 16 def. (9 EMD + AB, 7 EMD)	A predominant 1- to 2-wall component	Cortical autogenous bone particles were harvested from the buccal cortical plate by means of a bone scraper. The bone graft was collected from the surgical site adjacent to the intraosseous defect.
Yilmaz et al., 2010 [22]	RCT (Pa)	40/40 24/16	(30–50 years)	Test EMD + AB (20)	Ctr EMD (20)	Max: 18 def. (8 EMD + AB, 10 EMD) Mdb: 22 def. (12 EMD + AB, 10 EMD) IN/CA: 12 def. (6 EMD + AB, 6 EMD) PM: 14 def. (8 EMD + AB, 6 EMD) MO: 14 def. (6 EMD + AB, 8 EMD)	2 walls: 15 defects (7 EMD + AB, 8 EMD) 2-3 wall: 25 defects (13 EMD + AB, 12 EMD)	Cortico-cancellous autogenous bone was harvested from the retromolar area using a trephine bur with a diameter of 3 mm.
Agrali et al., 2016 [23]	RCT (Pa)	12/30 6/6	44.17 ± 7.80	Test EMD + AB (10)EMD (10)	Ctr OFD (10)	Max: NR Mdb: NR IN/CA: 10 def. (5 EMD + AB, 2 EMD alone, 3 OFD) PM: 10 def. (2 EMD + AB, 6 EMD alone, 2 OFD) MO: 10 def. (3 EMD + AB, 2 EMD alone, 5 OFD)	1-walled: 6 def. (1 EMD + AB, 4 EMD alone, 1 OFD) 1-2-walled: 20 def. (6 EMD + AB, 5 EMD alone, 9 OFD) 1-2-3-walled: 4 def. (3 EMD + AB, 1 EMD alone)	Autogenous bone was obtained from adjacent bone surfaces by using hand instruments Ochsenbein Periodontal Chisel CO_2_, Rhodes Back Action Periodontal Chisel C36/37, Hu-Friedy Inst. Co., Chicago, IL, USA).

RCT, randomized controlled trial; Pa, parallel group; m, months; w, weeks; M, male; F, female; MO, molars; PM, Premolars; CA, canines; IN, incisors; Mdb, mandible; Max, maxilla; def, defects; Ctr, control; EMD, enamel matrix derivative; AB, autogenous bone; OFD, open flap debridement; NR, not reported.

**Table 2 materials-12-02634-t002:** Main characteristics of selected studies: aim, inclusion criteria, and surgical protocols.

Study	Aim	Inclusion Criteria	Surgical Protocol
Guida et al., 2007 [18]	Assess the additional clinical benefit of autogenous cortical bone particles when added to EMD, compared to EMD alone, in the treatment of deep periodontal intraosseous defects.	1) No systemic diseases that contraindicated periodontal surgery; 2) No medications affecting periodontal status; 3) No pregnancy or lactation; 4) At least one intraosseous defect in need of surgical treatment after initial periodontal treatment and revaluation; 5) PD ≥ 6 mm; 3) Radiographic intraosseous defect ≥4 mm. 6) No full-mouth plaque score and full-mouth bleeding score >20% at the time of surgical procedure. Furthermore, third molars, teeth with Class III mobility, furcation involvement, inadequate endodontic treatment, or restoration were excluded.	After flap reflection, all soft tissue was removed from the defect, and the root surface was scaled and planed with hand and ultrasonic instruments. In all cases, the exposed root surfaces were conditioned with 24% EDTA gel for 2 min. The defect was then thoroughly rinsed with saline to remove gel remnants. For the EMD + AB group, particulate cortical bone was harvested from the buccal cortical plate by means of a bone scraper. A first layer of EMD was injected to condition the bone defect and the more apical portion of the root surface. AB was positioned to fill only the intrabony component of the defect. Finally, a second layer of EMD was injected to cover the grafted autogenous bone particles and to condition the portion of the root surface coronal to the bone crest. Therefore, a “sandwich” technique was adopted to treat the defect (i.e., apical layer of EMD, AB, and coronal layer of EMD). For the EMD group, the EMD gel alone was injected into the defect. Finally, flaps were positioned at the pre-surgery level or slightly coronal.
Yilmaz et al., 2010 [22]	Evaluate the healing of deep intrabony defects treated with either a combination EMD + AB or EMD alone	(1) No systemic diseases such as diabetes mellitus or cardiovascular diseases that could influence the outcome of the therapy; (2) No smokers; (3) A good level of oral hygiene (plaque index (PI) < 1); (4) Compliance with the maintenance programme; (5) Presence of one intrabony defect with a probing depth of at least 6 mm and an intrabony component of at least 3 mm, as detected on the radiographs.	Intracrevicular incisions were placed, and full-thickness flaps were raised vestibularly and orally. If necessary, vertical releasing incisions were performed. Following removal of granulation tissue from the defects, the roots were thoroughly scaled and planed using hand and ultrasonic instruments. In both groups, the root surfaces adjacent to the defects were conditioned for 2 min with an EDTA gel in order to remove the smear layer. The defects and the adjacent mucoperiosteal flaps were then thoroughly rinsed with sterile saline in order to remove all EDTA residues. Following root conditioning, EMD was applied to the root surfaces and into the defects with a sterile syringe. Cortico-cancellous AB was harvested from the retromolar area using a trephine bur. The remaining EMD was then mixed with AB and the defects were completely filled with the mixture of EMD + AB. Finally, the flaps were advanced coronally and closed with vertical or horizontal mattress sutures. The sites treated with EMD received exactly the same treatment, including root conditioning with EDTA, but without the application of AB.
Agrali et al., 2016 [23]	Evaluate the effects of EMD either alone or combined with AB applied to intrabony defects in chronic periodontitis patients on clinical/radiographic parameters and GCF TGF-β1 level and to compare with OFD.	(1) No systemic diseases that contraindicated periodontal surgery and could affect the consequences of the therapy; (2) No smoking; (3) No medications; (4) No pregnancy or lactation; (5) Good oral hygiene level (plaque index (PI) < 1) and full-mouth bleeding on probing score <20% after initial periodontal treatment (IPT); (6) Compliance with the maintenance program; (7) Minimum one intrabony defect existence with a probing depth (PD) ≥ 6 mm, radiographic depth ≥ 3 mm, as detected on radiographs.	After local anesthesia, sulcular incisions were made and full-thickness flaps were raised buccally and lingually, granulation tissues removed, and the root surfaces gently scaled and planed. In the EMD and combination groups, the exposed root surfaces were conditioned with 24% EDTA gel for 2 min. The surgical area was then rinsed with saline. EMD gel was injected onto the intrabony defects and root surfaces. Then, in the combination group, the adequate amount of AB obtained from adjacent bone surfaces by using hand instruments was mixed with the gel and placed into the bone defects. Finally, a second layer of EMD gel was injected to cover the AB. Then, the flaps were sutured.

EMD, enamel matrix derivative; AB, autogenous bone; PD, probing depth; EDTA, ethylenediaminetetraacetic acid; GCF, gingival crevicular fluid; TGF-β1, transforming growth factor-β1; OFD, open flap debridement.

**Table 3 materials-12-02634-t003:** Main characteristics of selected studies: outcomes, methods of evaluation, and conclusions.

Study	Time	Outcomes	Method of Evaluation	Conclusions
Guida et al., 2007 [18]	12 m	*Clinical parameters* -LPS (%) -LBS (%) -CAL (mm) -PD (mm) -REC (mm) *Intrasurgical parameters* -PBL (mm) -IBD (mm) *Radiographical parameters* -DEPTH (mm) -radiographic defect fill (percentage) -ANGLE (degrees)	-Periodontal probe (UNC 15, Hu-Friedy Mfg. Inc., Chicago, IL, USA). -A manual pressure-sensitive probe at approximately 0.3 N force with 1 mm increments). -LPS and LBS recorded dichotomously at surgical site as the presence or absence of supragingival plaque and bleeding on probing, respectively. -DEPTH, measured as the linear distance (in mm) -Radiographic defect fill (percentage) calculated as follows: (baseline DEPTH-12-month DEPTH)/baseline DEPTH 100 -ANGLE (degrees) the radiographic defect angle (ANGLE) at baseline, determined in degrees as the angle formed between the lines that represent the root surface of the involved tooth and the bone defect surface	Data support the clinical effectiveness of a regenerative procedure based on EMD application, either alone or in combination with a cortical AB, in the treatment of deep intraosseous defects without statistically significant differences. The combined EMD + AB procedure led to a reduced post-surgery recession and increased the proportion of defects with substantial clinical attachment level gain (≥6 mm).
Yilmaz et al., 2010 [22]	12 m	*Clinical parameters* -PI -GI -BOP (%) -PD (mm) -RAL (mm) -REC (mm) -PBL (mm) *Intrasurgical parameters* -INTRA (mm)	-Periodontal probe (UNC 15, Hu-Friedy Mfg. Inc, Chicago, IL, USA).	At 1 year after surgery, both therapies resulted in statistically significant clinical improvements compared with baseline, and although the combination of EMD + AB resulted in statistically significant higher soft and hard tissue improvements compared with treatment with EMD, the clinical relevance of this finding is unclear.
Agrali et al., 2016 [23]	6 m	*Clinical parameters* -PI -GI -BOP (%) -PD (mm) -RAL (mm) -REC (mm) *Intrasurgical measurements* -IDD (mm) *Radiographic parameters* -Bone fill (%) *Other parameters* -Determination of gingival crevicular fluid (GCF) transforming growth factor-β1 levels	-Periodontal probe (UNC 15, Hu-Friedy, Chicago, IL, USA) using an adapted acrylic stent with reference holes. -Long cone paralleling technique using an appropriate screening device (RWT Roentgenographic-System, Kentzler-Kaschner Dental GmbH, Germany). -GCF samples were collected with paper strips (PerioPaper® Oraflow Inc., New York, USA) just before surgery and 7, 14, 30, 90, 180 days after surgery and evaluated by enzyme-linked immunosorbent assay using a commercially available kit for TGF-β1 (Quantikine Human TGF-β1, R&D Systems, Inc., Minneapolis, MN, USA)	All treatment procedures led to significant improvements at 6 months (*p* < 0.01). Gain in attachment level (*p* < 0.01) and radiographic defect fill (*p* < 0.05) of the combination and EMD groups were found to be significantly higher than those of the control group, while the use of EMD either with AB or alone was observed to produce significantly less recession than the OFD (*p* < 0.05).

m, months; LPS, local plaque score; LBS, local bleeding score; CAL, clinical attachment level; PD, probing depth; REC, gingival recession; PBL, probing bone level; IBD, intrabony component of the defect; DEPTH, radiographic depth of the defect; ANGLE, radiographic defect angle; EMD, enamel matrix derivative; AB, autogenous bone; PI, plaque index; GI, gingival index; BOP, bleeding on probing; RAL, relative attachment level; FMPS, full mouth plaque score; INTRA, depth of the intrabony component of the defect; IDD, intrabony defect depth.

**Table 4 materials-12-02634-t004:** Quality assessment of the included studies.

Validity	Quality Criteria	Guida et al., 2007 [18]	Yilmaz et al., 2010 [22]	Agrali et al., 2016 [23]
External	Declared the use of specific protocol guidelines	no	no	no
	Representative population group	yes	yes	yes
	Eligibility criteria defined	yes	yes	yes
Internal	Consecutive enrollment	yes	yes	yes
	Random allocation	yes	yes	yes
	Allocation concealment	NR	yes	NR
	Blinding of the patient	NA	NA	NA
	Blinding of the examiner	yes	yes	no
	Blinding of the statistician	NR	NR	NR
	Reported loss to follow-up	yes	yes	yes
	No. (%) of dropouts	0	0	0
	Treatment identical, except for intervention	yes	yes	yes
Statistical	Sample size calculation and power	yes	yes	yes
	Point estimates presented for primary outcome	yes	yes	yes
	Measures of variability for the primary outcome	yes	yes	yes
	Intention to treat analysis	NR	NR	NR
	Coherent data presentation	yes	yes	yes
Clinical aspects	Study design	RCT parallel	RCT parallel	RCT parallel
	Validated measurement	yes	yes	yes
	Calibration of examiner	yes	yes	yes
	Estimated potential risk of bias	Low	Low	Low

NR, not reported; NA, not applicable; RCT, randomized controlled trial.

**Table 5 materials-12-02634-t005:** Clinical characteristics of intrabony defects at baseline.

Authors	PD (mm)	CAL/RAL (mm)	REC (mm)	PI	GI	BOP (%)	LPS (%)	LBS (%)
Guida et al., 2007 [18]	EMD (9.6 ± 1.7) EMD + AB (9.1 ± 1.6)	EMD (10.6 ± 1.3) EMD + AB (10.3 ± 1.5)	EMD (1.1 ± 1.0) EMD + AB (1.1 ± 0.9)	NA	NA	NA	EMD (21.4) EMD + AB (21.4)	EMD (71.4) EMD + AB (50.0)
Yilmaz et al., 2010 [22]	EMD (8.2 ± 0.7) EMD + AB (8.4 ± 1.2)	EMD (11.3 ± 0.9) EMD + AB (11.7 ± 1.0)	EMD (3.1 ± 1.1) EMD + AB (3.3 ± 1.5)	EMD (0.4 ± 0.1) EMD + AB (0.5 ± 0.1)	EMD (1.3 ± 0.3) EMD + AB (1.2 ± 0.2)	EMD (49.00) EMD + AB (50.00)	NA	NA
Agrali et al., 2016 [23]	EMD (8.30 ± 1.70) EMD + AB (7.93 ± 1.66) OFD (7.60 ± 1.51)	EMD (13.70 ± 2.58) EMD + AB (13.06 ± 1.77) OFD (12.10 ± 2.13)	EMD (5.40 ± 1.96) EMD + AB (5.12 ± 1.91) OFD (4.70 ± 1.70)	EMD (0.65 ± 0.24) EMD + AB (0.55 ± 0.16) OFD (0.75 ± 0.26)	EMD (0.90 ± 0.21) EMD + AB (0.78 ± 0.24) OFD (0.90 ± 0.21)	EMD (55.00 ± 10.54) EMD + AB (60.33 ± 17.29) OFD (62.50 ± 17.68)	NA	NA

PD, probing depth; CAL, clinical attachment level; RAL, relative attachment level; REC, gingival recession; PI, plaque index; GI, gingival index; BOP, bleeding on probing; LPS, local plaque score; LBS, local bleeding score; EMD, enamel matrix derivative; AB, autogenous bone; NA, not available.

**Table 6 materials-12-02634-t006:** Radiographic and intrasurgical characteristics of intrabony defects at baseline.

Authors	Radiographic Parameters		Intrasurgical Parameters	
	DEPTH (mm)	ANGLE (degrees)	PBL (mm)	IBD/IDD/INTRA (mm)
Guida et al., 2007 [18]	EMD (6.5 ± 2.9) EMD + AB (6.5 ± 1.8)	EMD (31.5 ± 2.4) EMD + AB (30.9 ± 12.6)	EMD (11.7 ± 1.7) EMD + AB (10.9 ± 2.0)	IBD EMD (6.2 ± 2.0) EMD + AB (7.0 ± 1.2)
Yilmaz et al., 2010 [22]	NA	NA	EMD (12.1 ± 0.9) EMD + AB (12.3 ± 1.0)	INTRA EMD (5.2 ± 0.7) EMD + AB (5.4 ± 1.0)
Agrali et al., 2016 [23]	NA	NA	NA	IDD EMD (6.40 ± 1.95) EMD + AB (5.20 ± 1.39) OFD (5.60 ± 1.64)

DEPTH, radiographic depth of the defect; ANGLE, radiographic defect angle; PBL, probing bone level; IBD, intraosseous component of the defect; IDD, intrabony defect depth; INTRA, the depth on the intrabony component; EMD, enamel matrix derivative; AB, autogenous bone; NA, not available.

**Table 7 materials-12-02634-t007:** Changes in BOP, PD, CAL, REC, RAL, DEPTH, and bone fill.

Authors	BOP (%)	PD (mm)	CAL/RAL (mm)	REC (mm)	DEPTH (mm)	BONE FILL (%)
Guida et al., 2007 [18]	NA	EMD (5.6 ± 1.7) EMD + AB (5.1 ± 1.7) NS	EMD (4.6 ± 1.3) EMD + AB (4.9 ± 1.8) NS CALgain < 2 mm EMD (0%) EMD + AB (0%) CALgain 2–3 mm EMD (21%) EMD + AB (29%) CALgain 4–5 mm EMD (57%) EMD + AB (21%) CALgain ≥ 6 mm EMD (21%) EMD + AB (50%)	EMD (1.1 ± 0.7) EMD + AB (0.3 ± 0.8) *	EMD (4.3 ± 2.4) EMD + AB (4.3 ± 1.3) NS DEPTH gain < 2 mm EMD (7%) EMD + AB (0%) DEPTH gain 2–3mm EMD (29%) EMD + AB (43%) DEPTH gain 4–5 mm EMD (50%) EMD + AB (43%) DEPTH gain ≥ 6 mm EMD (14%) EMD + AB (14%)	EMD (64.8 ± 24.1) EMD + AB (68 ± 17.3) NS
Yilmaz et al., 2010 [22]	EMD (16.00); EMD + AB (15.00)	EMD (4.6 ± 0.4) EMD + AB (5.6 ± 0.9) *	EMD (3.4 ± 0.8) EMD + AB (4.2 ± 1.1) * RALgain < 2 mm EMD (0%) EMD + AB (0%) RALgain 2–3 mm EMD (35%) EMD + AB (10%) RALgain 4–5 mm EMD (55%) EMD + AB (85%) RALgain 6 mm EMD (0%) EMD + AB (5%)	EMD (1.2 ± 0.8) EMD + AB (1.4 ± 0.9) NS	NA	NA
Agrali et al., 2016 [23]	EMD (42.50 ± 12.08) EMD + AB (47.71 ± 18.49) OFD (25.00 ± 0.00)	EMD (5.00 ± 1.41) EMD + AB (4.71 ± 1.63) OFD (4.40 ± 1.17)	EMD (4.50 ± 3.24) EMD + AB (3.55 ± 1.46) OFD (1.60 ± 0.70)	EMD (−0.50 ± 2.72) EMD + AB (−1.16 ± 1.62) OFD (−2.70 ± 0.95)	NA	EMD (65.98% ± 14.76%) EMD + AB (64.56% ± 24.23%) OFD (35.31% ± 20.56%)

NA, not available; NS: not significant; *: *p* < 0.05.

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
