# Peer review of "Enamel Matrix Derivative and Autogenous Bone Graft for Periodontal Regeneration of Intrabony Defects in Humans: A Systematic Review and Meta-Analysis"

_materials, 2019, doi:10.3390/ma12162634_

Round 1

Reviewer 1 Report

Dear authors

It is necessary to revise the list of references. It misses 12 references in the list : 11 in the introduction and one in table 2.

Choose fig or Fig

Replace et Al by et al

Correct in the text Yilmaz instead of Ylmaz

There are points in the middle of certain phrases to remove them

Review the text because sometimes there are too many intervals and sometimes it misses

In the text and tables choose min or minutes

You use different abbreviations (AB, ACB, ABG, ACBP) can you explain the differences or they represent the same thing?

There are many typing errors to correct

lanes 42 and 47: oral instead of horal

lane 50 obtain instead of abtain

lane 54 these instead of this

lane 58 technique instead of thecnique

lane 63  participates instead of partecipates

lane 64 in vitro in italics

lanes 71-75 the sentence is too long to cut it in half for better understanding

lane 74 eliminate J Int Clin Dent Res Organ

lanes 142, 239, 240  χ²

lane 158 males instead of male

lane 175 protocols instead of protocol

Table 3 millimeters instead of millimetres

lane 225 you mention a figure 4 that does not exist in your document???

lane 297 why autogenous bone in capitals while in line 293 in lower case?

lane 298 you mean supracrestal instead of sovracrestal ??? can you explain what this represents?

Author Response

Reviewer 1

It is necessary to revise the list of references. It misses 12 references in the list : 11 in the introduction and one in table 2.

Response: Missing references were added

Choose fig or Fig

Response:  “Fig.” was used everywhere in the text

Replace et Al by et al

Response:  “et Al.” was replaced by “et al.” everywhere in the text

Correct in the text Yilmaz instead of Ylmaz

Response: “Ylmaz” has been changed to “Yilmaz”

There are points in the middle of certain phrases to remove them

Response: points have been removed

Review the text because sometimes there are too many intervals and sometimes it misses

Response: The text has been reviewed and intervals corrected

In the text and tables choose min or minutes

Response: “min” has been preferred to “minutes” and changed everywhere in the text, accordingly

You use different abbreviations (AB, ACB, ABG, ACBP) can you explain the differences or they represent the same thing?

Response: the form “AB”, acronym of “autogenous bone” was used everywhere in the text and the other forms changed, accordingly.

There are many typing errors to correct

Response: a complete further control of the text has been done.

lanes 42 and 47: oral instead of horal

Response: the correction has been done

lane 50 obtain instead of abtain

Response: the correction has been done

lane 54 these instead of this

Response: the correction has been done

lane 58 technique instead of thecnique

Response: the correction has been done

lane 63  participates instead of partecipates

Response: the correction has been done

lane 64 in vitro in italics

Response: the correction has been done

lanes 71-75 the sentence is too long to cut it in half for better understanding

Response: the sentence  has been reduced, as suggested.

lane 74 eliminate J Int Clin Dent Res Organ

Response: the journal abbreviation has been deleted

lanes 142, 239, 240  χ²

Response: the correction has been done

lane 158 males instead of male

Response: the correction has been done

lane 175 protocols instead of protocol

Response: the correction has been done (l. 181)

Table 3 millimeters instead of millimetres

Response: the correction has been done

lane 225 you mention a figure 4 that does not exist in your document???

Response: the figure 4 has been added. It was lacking in the manuscript for a mistake. Sorry for this.

lane 297 why autogenous bone in capitals while in line 293 in lower case?

Response: the correction has been done

lane 298 you mean supracrestal instead of sovracrestal ??? can you explain what this represents?

Response: “sovracrestal” has been chamged to “supracrestal soft tissues”

Reviewer 2 Report

This manuscript untitled “Enamel matrix derivative and autogenous bone graft 2 for periodontal regeneration of infrabony defects in 3 humans. A systematic review and meta-analysis” were to assess the efficacy of EMD in combination with autogenous bone graft compared with the use of EMD alone for the regeneration of periodontal infrabony defects by systematic review and meta-analysis.
This aim of this paper is quite interesting, but the theme is so specific that it has limited in too many articles it has found. There is a very similar literature review, but with EDM and bone material substituents (Matarasso, 2015). In your case, of the three studies, it includes, two were already present in the previous systematic review, and you only added one article. And you could add just one more recent article (2016). Seems to be the great weakness of the article.
Generally, there are grammatical errors in this manuscript, but page 1 line 42 “horal”-oral.
The introduction section is well written. The methodology is adequate for the objective of the authors for this study. These evaluation methods are very specific and objective, However, the authors refer that the exclusion criteria will be explained in appendix 1. Where is appendix1?
It would be important to have access to this information.
Aim of the study corresponds to its conclusion. Results of the study induce this conclusion.
References should be standardized, as they are deformed as ref 2 and page 1 line 74  (Kataria) this author do not appear on references.

Author Response

Reviewer 2

This manuscript untitled “Enamel matrix derivative and autogenous bone graft 2 for periodontal regeneration of infrabony defects in 3 humans. A systematic review and meta-analysis” were to assess the efficacy of EMD in combination with autogenous bone graft compared with the use of EMD alone for the regeneration of periodontal infrabony defects by systematic review and meta-analysis. 
This aim of this paper is quite interesting, but the theme is so specific that it has limited in too many articles it has found. There is a very similar literature review, but with EDM and bone material substituents (Matarasso, 2015). In your case, of the three studies, it includes, two were already present in the previous systematic review, and you only added one article. And you could add just one more recent article (2016). Seems to be the great weakness of the article.
Generally, there are grammatical errors in this manuscript, but page 1 line 42 “horal”-oral.

Response: a complete revision of the text for grammatical errors has been done.

The introduction section is well written. The methodology is adequate for the objective of the authors for this study. These evaluation methods are very specific and objective, However, the authors refer that the exclusion criteria will be explained in appendix 1. Where is appendix1? 
It would be important to have access to this information.

Response: the sentence was deleted from the text. A list of the excluded articles with the reason is available, if requested.

Aim of the study corresponds to its conclusion. Results of the study induce this conclusion. 
References should be standardized, as they are deformed as ref 2 and page 1 line 74  (Kataria) this author do not appear on references.

Response: references were standardized and corrected as suggested.